# Exploring the Relationships between Key Ecological Indicators to Improve Natural Conservation Planning at Different Scales

**Lu Zhang** and **Zhiyun Ouyang ***

State Key Laboratory of Urban and Regional Ecology, Research Center for Eco-Environmental Sciences, Chinese Academy of Sciences, Beijing 100085, China; luzhang@rcees.ac.cn
* Correspondence: zyouyang@rcees.ac.cn; Tel.: +86-010-6284-9191

**Abstract:** Biodiversity, regulating ecosystem services (RES), and vegetation productivity are key indicators to instruct natural conservation planning. Decision makers often hope that ecosystems can be protected by focusing on certain key indicators, which requires an understanding of the relationships between the indicators. Using individual case studies, many have argued that these indicators commonly have significant relationships. However, these relationships at different spatial scales are unclear. Therefore, in this study, biodiversity and ecosystem services are modelled by the ecological niche model, the universal soil loss equation, and the equation of water balance in two study areas at different scales. The influence of vegetation productivity on the spatial pattern of other ecological indicators in the two areas is examined by a spatial lag model. The contributions of the driving factors on biodiversity distribution at both scales are identified by a boosted regression tree (BRT) model. The results showed that at the fine scale, the spatial correlations were strongest for species richness, especially mammalian species richness, and water retention. However, biodiversity had no significant relationship with vegetation productivity. In contrast, at a coarser scale, the correlation was stronger between plant diversity and regulating ecosystem services. In addition, plant diversity was significantly correlated with vegetation productivity. These differences between scales were controlled by various explanatory variables. At the fine scale, biophysical and climatic factors had the strongest effects on biodiversity distribution, while Net Primary Productivity (NPP) and ecoregion also had relatively high influences on biodiversity at the coarse scale. This demonstrates the critical importance of spatial scale in selecting conservation indicators. We suggest that rare mammalian species richness or flagship mammal species are suitable as conservation surrogates in fine-scale conservation planning. However, at a coarser scale, selecting vegetation patches with more rare plant species and high productivity for each ecoregion is a workable alternative method for conservation planning.

**Keywords:** biodiversity; regulating ecosystem service; vegetation productivity; conservation indicator; scale

## 1. Introduction

Natural ecosystems directly or indirectly provide essential services and functions for human survival [1]. However, in the past century, the intensity of human activities has increased unprecedentedly. Factors such as climate change, land use change, species invasions, and the spread of infectious diseases have seriously threatened ecosystems and their associated species [2,3]. To effectively protect natural ecosystems, conservation areas have been established worldwide. Conservation areas are often established by restricting local economic development. Appropriate conservation planning can solve the problems of space allocation and management of conservation resources while taking into account

natural, social, and economic characteristics. However, ecosystems are complex and diverse. The choice of indicators as proxies of ecosystem importance is crucial for improving the effectiveness and viability of protected areas and networks [4,5].

To build conservation networks, many approaches have been developed, such as ecosystem zonation, gap analysis, and systematic conservation planning. In processing these methods, the first step is to define the final ecological proxy among the various ecological indicators that must be protected. These proxies will be used as the core factors for planning, evaluation, or site selection [6]. Biodiversity maintenance is an important function of natural ecosystems; it has been used in developing targets for protection in multiple regions worldwide and is associated with many ecological functions, such as water purification, hydrological regulation, and pollination [7,8]. Biodiversity includes the three aspects of genetic, species, and ecosystem biodiversity, where genetic diversity is the finest level of biodiversity measurement. The measured results for genetic diversity are the most intuitive argument for demonstrating the evolutionary potential of a species [9]. The use of genetic diversity as an indicator of biodiversity ensures that protected areas include regions that have the greatest potential for the evolution of a species [10]. This method has high certainty and is suitable for planning at a fine scale. However, it is seldom used in regional conservation planning due to the difficulty of implementation, high cost, and complexity of the time-space selection for sampling at coarse scales, such as first-class basins, geographical regions, and mountain regions.

In contrast, species richness is easier to understand than genetic diversity, and it is easier to collect data related to it for regional conservation planning. Commonly, conservationists tend to select several kinds of flagship species within an ecosystem as proxies for conservation planning, hoping that other ecological indicators can be protected when the habitats of these species are protected [11,12]. However, in some cases, the habitats of these flagship species do not represent the ecological importance of the region. Therefore, some other ecological indicators should be included depending on the characteristics of the planning region [13], such as using ecosystem productivity as an alternative indicator for ecosystem conservation planning at a coarse scale [14]. For example, in marine ecosystems, the correlation of aquatic species distribution and basal productivity is extremely high, and species data are difficult to obtain. Therefore, basal productivity can be a good proxy for marine conservation planning [15]. Ideally, some works argue that species and vegetation productivity require simultaneous protection in a data-deficient area to achieve the conservation goal [16].

The interrelationship between biodiversity and ecosystem function depends on factors such as ecosystem type, climate, topography, soil, and human disturbance, which are closely correlated with biomass and ecosystem productivity [17,18]. It is not suitable to use all the indicators during the planning stage, which easily results in overprotection and reduces the applicability of the conservation plan. If the spatial relationships among the factors can be clarified, they can provide an important reference for ecological conservation planning and help to gain the synergy between biodiversity and regulating ecosystem services. However, the relationships might vary by spatial scale, and there are few comparative studies on this regime. Measurement and discussion of this issue will benefit mountain nature reserve planning.

In order to explore the covariance between RES, vegetation, and biodiversity at different scales and identify the scale differences at which spatial overlap acts and measure the contribution of determinants impacting biodiversity distributions at both scale, we performed the analysis based on mapping of water retention, soil retention (RES), NPP, biomass (ecosystem productivity), mammal, bird, plant, and sum of all indicator species (biodiversity). The covariance analysis begins by spatial intersection and correlation between biodiversity and RES, followed by a spatial lag model testing the relationships between biodiversity and vegetation productivity. Furthermore, we employed a Boost Regression Tree model (BRT) to estimate biophysical factors driving biodiversity distributions at both scales. Therefore, the objectives of this study are (1) to measure the correlations of biodiversity, ecosystem functions, and ecosystem productivity and (2) to identify the key factors impacting the biodiversity distribution between scales.

## 2. Methods

### 2.1. Study Areas

The study area at the coarse scale includes mainland China and Hainan Island, with an area of 9.45 million km$^2$. At the fine scale, the Qinling Mountain region is located in Shaanxi Province in central China between 32° N and 34° N and has an area of 57,800 km$^2$ (Figure 1). This region is regarded as the natural dividing line between China's north and south climatic zones and encompasses the watershed of the Yangtze River and Yellow River. It is also a transitional zone between the subtropical zone and the warm temperature zone, keeping core habitats for rare and endangered species, such as the giant panda (*Ailuropoda melanoleuca*) and the Sichuan golden snub-nosed monkey (*Rhinopithecus roxellana*), and providing key water retention functions for the Yangtze River and Yellow River. The typical evergreen and deciduous broad-leaved forest and temperate mixed forest ecosystem of the Qinling Mountains is a very representative trait of forest ecosystems in central and southern China.

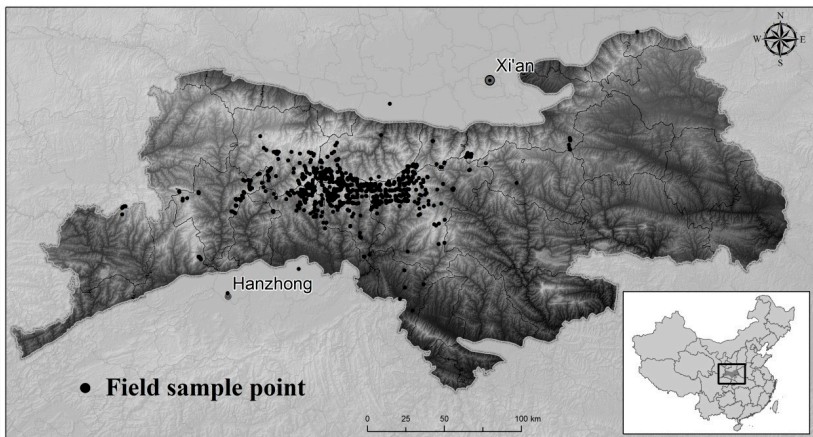

**Figure 1.** Spatial location of the Qinling Mountain region and survey sample points for biodiversity.

Data Acquisition

The biodiversity data for the Qinling mountain region were obtained from a line transect survey of wild animals and plants from 2007–2013 (Figure 1), including 23 indicator species (Supplementary Material S1) decided by the International Union for Conservation of Nature (IUCN) species Red List [19] and the Chinese Animal and Plant Species Protection List [20]. There were 1439 indicator species, including species on the IUCN Red List for the critically endangered (CR) and endangered (EN) categories and the protected species on the Chinese Species Protection List at the first level.

The data sources for the habitat information included the International Union for Conservation of Nature [19] and the Scientific Database of China Plant Species [21]. Land use/land cover data were from the China Cover dataset (30 m resolution) for the simulation of biodiversity and ecosystem services. The normalized difference vegetation index (NDVI) data were obtained from the MODIS 13Q1 product with 250 m resolution (National Aeronautics and Space Administration, Washington, DC, USA). The topographic data were obtained from the GDEM (Global Digital Elevation Model was developed jointly by the U.S. National Aeronautics and Space Administration, Washington, DC, USA and Japan's Ministry of Economy, Trade, and Industry, Tokyo, Japan) mosaic (30 m). The precipitation and temperature data were obtained from the National Weather Service (250 m).

### 2.2. Exploring the Relationships between Biodiversity, Ecosystem Services, and Vegetation Productivity

In the Qinling Mountain region, we used the maximum entropy (MAXENT) model [22] to simulate species distribution. The input layers included (1) a physical environment layer, including altitude, slope, aspect, distance to small rivers, and distance to large rivers; (2) a biological factor

layer that included broad-leaf forest, mixed forest, coniferous forest, shrub, meadow, and grassland; and (3) a human interference factor layer that included the density of the residential areas, distance to a main road, the distance to secondary roads, and the distance to farmland. In potential range of each indicator species, which decided by local monitoring reports and expert opinion, we made trace point survey of species along 381 line transect keeping 1 transect within $5 \times 5$ km$^2$ area. Points were not recorded within 150 m distance of other records of same species to reduce sampling bias. Seventy-five percent of the points were randomly selected for developing the model, and 25% of the points were used for the model validation by ROC analysis (Supplementary Material S3, Figures S1 to S7). The continuous results were changed to Boolean maps by a threshold searching method based on kappa analysis [23]. Spatial resolution of all feeding variables was resampled to 90 m.

At the national scale, 1423 indicator species were included for the biodiversity simulation. The ecological niche model was used for the habitat modelling (Supplementary Material S2). The universal soil loss equation was used for the soil retention mapping, and the water balance equation (Supplementary Material S3) was used for the water retention mapping (Supplementary Material S4).

The statistical correlations between the ecosystem services and the species richness were determined using the Spearman correlation test. Sample points of the vegetation types are randomly generated with a 1 km minimum distance between the points by Arcgis 10.0 software (ESRI Inc., RedLands, CA, USA). Then, the biodiversity distribution was divided into birds, mammals, and plants by using a spatial overlay analysis of the biodiversity data and the ecosystem services. First, the spatial intersection of the species richness and the ecosystem services was performed. Second, the data were classified into four levels of importance (L1: 1%–50%, L2: 51%–75%, L3: 76%–90% and L4: 91%–100%) by functional value (e.g., the number of species for the species richness mapping; tons for the water or soil retention service). To measure the impacts of vegetation productivity (explanatory variables) on species richness and regulating ecosystem services (dependent variables), we performed a test using a spatial lag model to control the spatial autocorrelation using GEODA software (Center for Spatial Data Science, University of Chicago, IL, USA). All the analyses were carried out based on two areas of the fine and coarse scales.

### 2.3. Identifying the Contributions of Correlated Factors on Biodiversity Distribution at Different Scales

The boosted regression tree (BRT) model was used to examine how species richness was affected by the regional climate, topography, soil, and potential biota. We chose the BRT because the model can help to achieve greater accuracy by handling many problematic issues of the feeding dataset, such as nonlinear relationships, missing data, and multicollinearity existing in both the dependent and independent observational data [24]. Dependent variables are numbers of indicator species at both scale. Independent variables include FC (%): Maximum forest cover during growing season; Biomass (gC/m$^2$): Aboveground biomass; NPP (gC/m$^2$): Annual Net primary productivity; DEM (m): Digital Elevation Model; Slope (°): Degree of slope; Temperature (°C): Annual temperature; Precipitation (mm): Annual precipitation; Radiation (kJ/cm$^2$): Annual radiation; SOM (gC/m$^2$): soil organic matter; Ecoregion (dimensionless): Identification numbers of ecoregions. In this study, we tested Variance Inflation Factor (VIF) along with a Least Squares Regression, and removed non-significant variables for feeding BRT model. Then, we built a BRT model using the R language package gbm [25] at two scales to identify the contribution of the explanatory factors on the distributions of the species richness using the recommended settings from Elith and Leathwick, 2008 [26]: family = "poisson", tree complexity = 5, learning rate = 0.005, bag fraction = 0.5.

## 3. Results

### 3.1. The Relationship between Biodiversity and Regulating Ecosystem Services

At the coarse spatial scale, there were significant positive correlations between soil retention and bird, mammal, and plant diversity ($p < 0.01$); the correlation was highest for soil retention and plant

diversity, with a correlation coefficient of 0.4 ($p < 0.01$). The distributions of the three biodiversity indicators were positively correlated with the water retention function ($p < 0.01$). Among them, the correlation of water retention and plant diversity was also the highest ($p < 0.01$). However, at the fine scale, soil retention had no significant relationship with any of the biodiversity indicators ($p > 0.01$). In contrast, water retention exhibited a significant positive correlation with the biodiversity indicators ($p < 0.01$). Furthermore, the correlation coefficients were higher at the fine scale than at the coarse scale, with the highest correlation coefficient of 0.605 ($p < 0.01$) for the mammal species richness (Table 1) (Supplementary Material S5 to see the spatial distribution for all the indicators.)

**Table 1.** Correlation coefficients of species richness and regulating ecosystem services in two case areas.

| Regions | Ecological Indicators | Soil Retention | Water Retention | Birds | Mammals | Plants |
|---|---|---|---|---|---|---|
| Qinling Mountain | Soil retention | - | 0.120 ** | 0.008 | 0.036 | 0.038 |
| (fine scale) | Water retention | 0.120 ** | - | 0.148 ** | 0.605 ** | 0.300 ** |
| China | Soil retention | - | 0.555 ** | 0.149 ** | 0.148 ** | 0.400 ** |
| (coarse scale) | Water retention | 0.555 ** | - | 0.122 ** | 0.156 ** | 0.389 ** |

** $p < 0.01$.

The spatial distribution between the regulating ecosystem services and the biodiversity indicators at different importance levels was compared. At the fine-scale, the spatial overlap area was highest for water retention and mammals. The sum of the overlapped areas was more than 40%. In particular, the top 10% of the important areas of the two indicators were 60% intersected (Figure 2a). When the research area was expanded to mainland China, the spatial correlations changed. The largest intersection area occurred for water retention and plant diversity. Moreover, the overlap area increased by increasing the importance of the two indicators (Figure 2b).

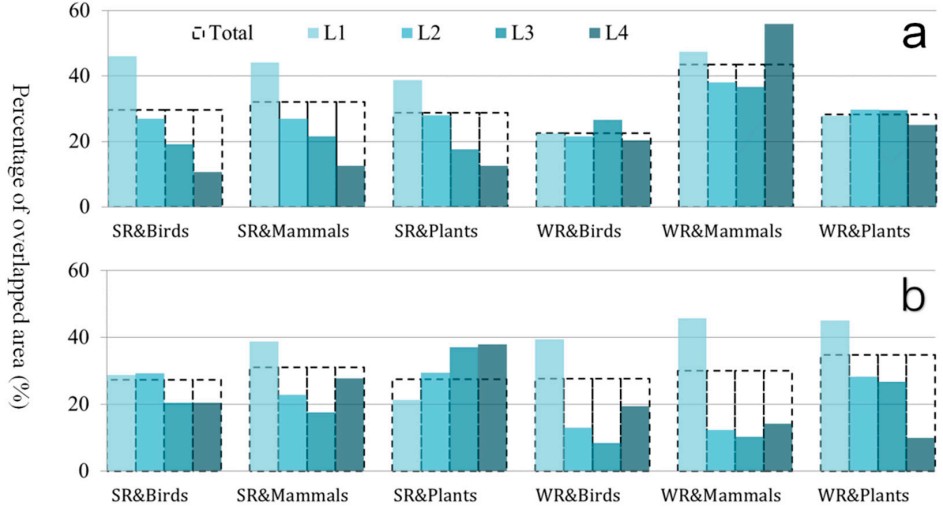

**Figure 2.** The spatial superposition relationship of ecosystem services and biodiversity: L4 represents the spatial overlap ratio of 10% of the most important regions for two elements. L3 represents the spatial overlap ratio of 76%–90% of the important regions for two elements. L4 represents the spatial overlap ratio of 51%–75% of the important regions for two elements. L1 represents the spatial overlap ratio of 1%–50% of the important regions for two elements. The total represents the spatial overlap of the region without level division. SR represents the soil conservation function. WR represents the water conservation function. (**a**) is the analysis at the fine scale and (**b**) is the analysis at the coarse scale.

*3.2. Impacts of Vegetation Productivity on Biodiversity and Regulating Ecosystem Services*

The results of the spatial lag model indicate that biomass had no significant influence on the spatial pattern of the two ecosystem services at the fine scale. NPP had no relationship with soil

retention and only a weak influence on water retention, with an $R^2$ of 0.053. In contrast, biomass and NPP showed a significant correlation with the biodiversity indicators. However, there was a negative correlation between NPP and plant diversity as well as between biomass and mammal diversity.

At the coarse scale, the correlation between vegetation productivity and ecosystem services, as well as biodiversity, was significant and higher than in the fine-scale case. It is clear that species richness increases with increasing vegetation productivity. In particular, a linear relationship with water retention was apparent, and the $R^2$ was 0.493. NPP also had significant linear correlations with ecological services and biomass (Table 2).

**Table 2.** Impacts of NPP and biomass on biodiversity and regulating ecosystem services.

| Fine Scale | NPP | | | Biomass | | | $R^2$ |
|---|---|---|---|---|---|---|---|
| | Coefficients | Std. Error | *p*-Value | Coefficients | Std. Error | *p*-Value | |
| Water retention | $-0.083$ | 0.009 | <0.001 | – | – | – | 0.053 |
| Soil retention | – | – | – | – | – | – | – |
| Bird richness | $2.148 \times 10^{-5}$ | $9.677 \times 10^{-6}$ | 0.027 | $3.656 \times 10^{-5}$ | $1.523 \times 10^{-5}$ | 0.016 | 0.006 |
| Mammal richness | $1.542 \times 10^{-5}$ | $2.379 \times 10^{-5}$ | <0.001 | $-8.614 \times 10^{-5}$ | $2.427 \times 10^{-5}$ | <0.001 | 0.023 |
| Plant richness | $-1.492 \times 10^{-5}$ | $7.152 \times 10^{-6}$ | 0.037 | $3.459 \times 10^{-5}$ | $1.126 \times 10^{-5}$ | 0.002 | 0.009 |
| **Coarse scale** | **NPP** | | | **Biomass** | | | $R^2$ |
| | Coefficients | Std. Error | *p*-Value | Coefficients | Std. Error | *p*-Value | |
| Water retention | 0.055 | 0.001 | <0.001 | 0.017 | 0.008 | 0.030 | 0.493 |
| Soil retention | 0.011 | $2.287 \times 10^{-4}$ | <0.001 | – | – | – | 0.260 |
| Bird richness | $6.921 \times 10^{-6}$ | $6.388 \times 10^{-7}$ | <0.001 | $4.046 \times 10^{-5}$ | $5.026 \times 10^{-6}$ | <0.001 | 0.095 |
| Mammal richness | $2.925 \times 10^{-5}$ | $1.770 \times 10^{-6}$ | <0.001 | $8.089 \times 10^{-5}$ | $1.392 \times 10^{-5}$ | <0.001 | 0.131 |
| Plant richness | $6.411 \times 10^{-5}$ | $1.872 \times 10^{-6}$ | <0.001 | – | – | – | 0.164 |

*3.3. Contributions of the Explanatory Variables on Biodiversity Distributions between Scales*

The different relationships between biodiversity, regulating ecosystem services, and vegetation productivity at the two scales largely depended on the fact that the biodiversity indicators were under various control factors between scales. We found all factors significantly impacted biodiversity distribution in coarse scale case, while NPP, Aspect, and Soil organic matter show non-significant effect on biodiversity at fine scale (Table 3). By contrast of variable contributions on biodiversity distribution, as shown in Figure 3, at the coarse scale (mainland China), NPP had the greatest influence on species richness (29.3%), followed by the DEM (22.4%), ecoregion (9.7%), and temperature (9.1%) (Figure 3). At the fine scale (Qinling Mountains), the DEM showed the strongest effects impacting species distribution (35.9%), followed by precipitation (23.7%), radiation (14.4%), and temperature (9.9%) (Figure 4).

**Table 3.** Results of Least Squares Regression collinearity test.

| Variable | Coefficients | | Std. Error | | *p*-Value | | VIF | | $R^2$ | |
|---|---|---|---|---|---|---|---|---|---|---|
| | CS | FS | CS | FS | CS | FS | CS | FS | CS | FS |
| Constant | 2.866 | $-43.178$ | 0.884 | 4.621 | | | | | | |
| Biomass | 0 | $-0.001$ | 0 | 0 | 0 | 0.032 | 2.175 | 1.165 | | |
| FC | $-0.203$ | 0.468 | 0.011 | 0.042 | 0 | 0 | 8.37 | 1.365 | | |
| NPP | 0 | 0.004 | 0 | 0.003 | 0 | 0.141 | 16.168 | 1.145 | | |
| DEM | 0.001 | 0.003 | 0 | 0.001 | 0 | 0 | 4.14 | 7.596 | | |
| Slope | 0.089 | 0.005 | 0.014 | 0.012 | 0 | 0.661 | 1.499 | 1.011 | 0.55 | 0.659 |
| Temperature | $-0.013$ | $-0.041$ | 0.003 | 0.012 | 0 | 0.001 | 6.65 | 6.011 | | |
| Precipitation | 0 | 0.012 | 0 | 0.001 | 0 | 0 | 5.132 | 1.649 | | |
| Radiation | 0 | $-0.007$ | 0 | 0.001 | 0.006 | 0 | 2.659 | 1.32 | | |
| SOM | 0.01 | 0 | 0.002 | 0.003 | 0 | 0.962 | 1.479 | 1.111 | | |
| Ecoregion | 0.053 | 0.063 | 0.007 | 0.015 | 0 | 0 | 1.334 | 1.256 | | |

CS: Coarse scale case; FS: Fine scale case; VIF: Variance Inflation Factor; FC (%): Maximum forest cover during growing season; Biomass (g/m$^2$): Aboveground biomass; NPP (gC/m$^2$): Annual Net primary productivity; DEM (m): Digital Elevation Model; Slope (°): Degree of slope; Temperature (°C): Annual temperature; Precipitation (mm): Annual precipitation; Radiation (kJ/cm$^2$): Annual radiation; SOM (gC/m$^2$): soil organic matter; Ecoregion (dimensionless): Identification numbers of ecoregions.

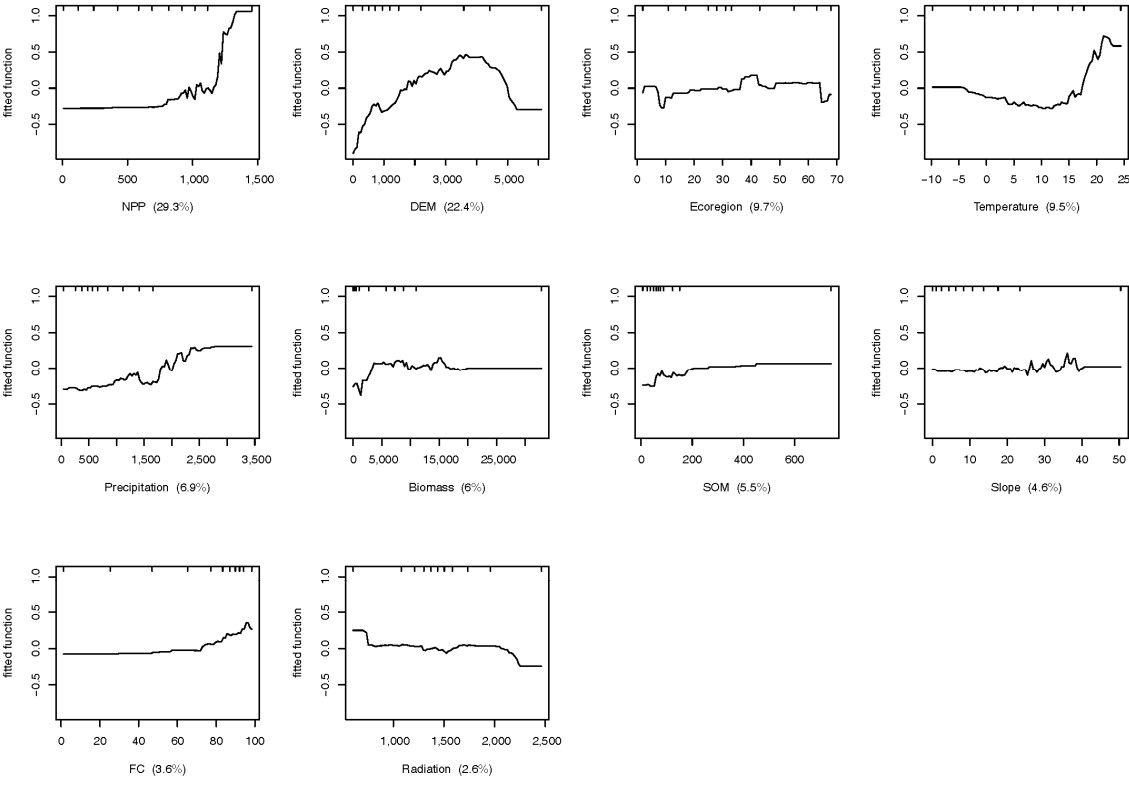

**Figure 3.** Contribution of impact factors on the distribution of species richness at the coarse scale.

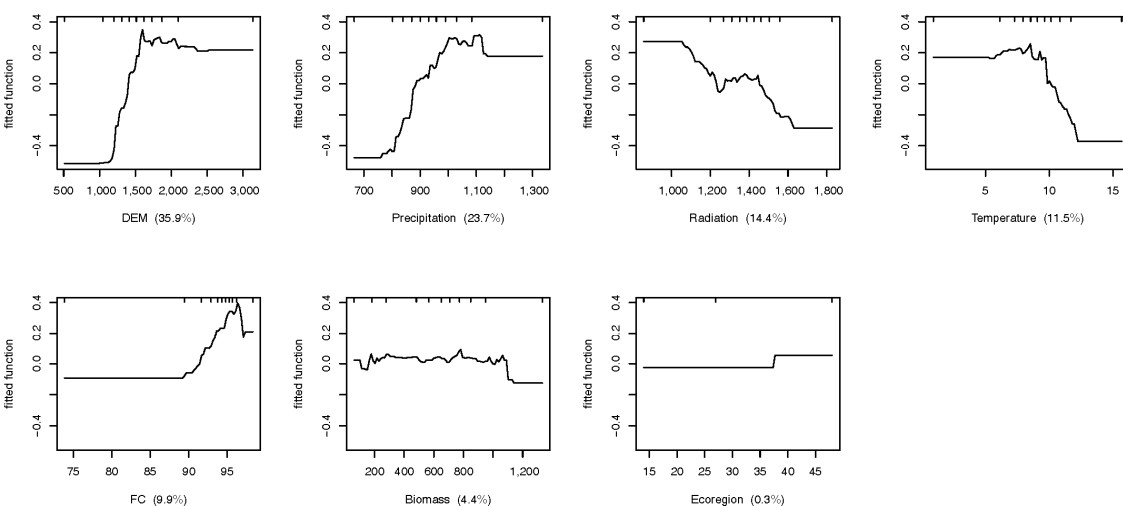

**Figure 4.** Contribution of impact factors on the distribution of species richness at the fine scale

## 4. Discussion

Understanding the spatial relationships between ecological indicators is important for answering the question of how to select suitable surrogates of ecological importance under limited conservation resources to improve the efficiency of ecological conservation planning [27,28]. However, this relationship is so complex that it depends on various impact factors, such as ecoregions and spatial scale [29,30]. Therefore, in this study, the relationship between biodiversity and regulating ecosystem services was determined using long-term data in areas with different scales. We chose soil retention and water retention for the regulating ecosystem services, which is a limited component of ecosystem service content. In China's major conservation policies, such as China's ecological function zoning and ecological redline policy, there are only four types of indicators, water retention, soil retention,

biodiversity protection, and sand storm prevention, to identify conservation priority areas. Among these indicators, the spatial distribution of sand storm prevention, which has not been considered in this research, has a sharp distinction between the other indicators.

According to the analysis, there was a positive correlation between biodiversity and ecosystem services. Areas with higher biodiversity provide more ecosystem services, which is consistent with many other studies [31–33]. However, some relationships between the ecological indicators differed at the two scales. At the coarse scale, the correlation was apparent between soil retention and water retention and was also significantly correlated with the richness of birds, mammals, and plants. Among them, the correlation was highest for plant richness. The results further demonstrated that the percentage of overlapping areas between the most important 10% of the areas regulating ecosystem services and plant richness was higher than in other areas. This indicates that plant diversity has the ability to express other ecological indicators on a large scale. In the Qinling Mountain region, there was no significant relationship between soil retention and species richness. However, the relationship between water retention and biodiversity indicators was apparent. In particular, the overlap ratio was nearly 60% between mammal richness and water retention in the top 10% important areas. This demonstrates that in a similar forest ecosystem to the Qinling Mountains, conservation planning for water resources in the forest would be able to cover the core habitats of rare mammal species as well.

Various ecological mechanisms between scales can partly explain the differences in the results for the two cases. Qinling is one of the most biodiverse regions with high forest coverage in China and is also the core habitat for the giant panda and the Sichuan golden monkey. Sufficient habitat makes wild mammals select preferable forest patches with enough food, water, and further distances from human disturbances as their daily habitat [34]. Therefore, at the fine scale, the overlap was high for regions of water conservation and mammal diversity. However, the situation differs nationally for other vegetation classes. For example, the forested areas in north-eastern China have higher water conservation functions, but the mammalian richness is lower than in the mountain regions of central and southern China. Therefore, the relationship was not significant at the national scale [35]. In contrast to mammals, plant diversity is more averagely distributed in different forested regions. Therefore, it was significantly correlated with ecological services. The method chosen for biodiversity estimation to assess differences may also be a factor that impacted the result. In the Qinling Mountains, we considered the frequency of different species for the same forested area to identify the core habitat using the Maxent model, which is hardly used in a large region study because of data availability. This analysis design matches the goals of conservation policy in China at different scales [36]. For conservation work at the national scale, decision makers only need to select key mountains, ecoregions, or administrative districts with important regulating ecosystem services. At the fine scale, a clear boundary within a similar ecosystem is required. Therefore, the conclusions of this study integrated with the modelling method can be used to guide conservation planning at different scales in China.

Vegetation biomass and NPP are important indicators for estimating ecosystem functions, and some studies have demonstrated that ecosystem productivity is a key driving factor of biodiversity and ecosystem services [37,38]. Moreover, there are many advanced methods and data sources for the monitoring of biomass and NPP [39]. Therefore, studies on the relationship between these two factors and biodiversity and ecological services can help to simplify the determination of the ecological indicators for conservation. The regression analysis used in this study demonstrates that the influence of vegetation productivity on regulating ecosystem services is significant at the coarse scale. Regions with high biomass and high productivity also have high water retention and biodiversity values. When the conservation target is set for protecting high vegetation productivity, areas with high regulating services and biodiversity can also be included at a large scale. However, at the fine scale, this relationship is very ambiguous. The reasons for this difference depend on various factors. At a coarse scale, biodiversity distribution is decided by discrepancies between ecoregions with different altitudes, ecosystem types, and climates. These factors also control the NPP distribution at a large scale. Therefore, the BRT analysis results showed a significant spatial relationship between

NPP and biodiversity. While at the fine scale, similar factors control the macroscopic patterns of biodiversity, the habitat selection by species is controlled primarily by biophysical and climatic factors [40,41]. However, the correlation between these factors and NPP is low because of the similar macroscopic biogeography conditions [42]. Therefore, at present, the interrelationship between vegetation productivity, biodiversity, and regulating ecosystem services is not clear. If this indicator is used for determining the final protection indicator, it is possible to miss important information crucial for protection.

## 5. Conclusions

This study shows that the spatial scale of conservation planning should be taken into full account for the determination of ecological indicators for conservation planning. The spatial overlap and correlation analysis demonstrated that in a mountain region with a high value of biodiversity sustainability and regulating ecosystem services, mammal species richness is a core indicator for establishing a conservation network which are mainly driven by biophysical and climatic factors. At a coarser spatial scale, plant species richness becomes more important to benefit synergy of ecosystem functions. At national scale, NPP and ecoregion also have high influences in addition to biophysical and climatic factors on the pattern of species richness.

**Supplementary Materials:** The following are available online at http://www.mdpi.com/1999-4907/10/1/32/s1, Supplementary Material S1: Selection of indicator species in Qingling mountain study, Supplementary Material S2: Biodiversity mapping in mainland China study, Supplementary Material S3: Examples of ROC curve verification of mammal indicator species in Qinling Mountain, Supplementary Material S4: Modeling of soil retention, Supplementary Material S5: Mapping of ecological indicators.

**Author Contributions:** Conceptualization, Z.O. and L.Z.; Methodology, L.Z.; Software, L.Z.; Validation, Z.O. and L.Z.; Formal Analysis, L.Z.; Investigation, L.Z.; Resources, L.Z.; Writing—Original Draft Preparation, L.Z.; Writing—Review & Editing, Z.O.; Supervision, Z.O.; Project Administration, L.Z.; Funding Acquisition, L.Z.

**Funding:** Our study was funded by the China Natural Science Foundation (71603251).

**Conflicts of Interest:** The authors declare no conflict of interest.

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
