# Peer review of "Exploring the Relationships between Key Ecological Indicators to Improve Natural Conservation Planning at Different Scales"

_forests, doi:10.3390/f10010032_

Reviewer 1 Report

Dear authors,

I recommend that you formulate a relevant research question, objective or hypothesis, design a study to address those, search and analyze appropriate data, and draw valid conclusions. I also recommend that you address biodiversity, indicators and other basic terms and concepts correctly. The sentence "Biodiversity is an important indicator for measuring the importance of natural ecosystems ..." (l 54-55) clearly shows a serious misunderstanding of key foundations. Your conclusions are not supported, by data, method or study design, or by a combination of these. Your main conclusions that mammal species richness is an indicator for conservation, that NNP and biomass influence biodiversity and ecological functions, and that scale is a dependent variable, are very superficial and can be found in any basic text book in biology and ecology. Also, other variables ("latent", l. 282) for sure are both numerous and very important to consider. The fact that you chosed the term "might"  (l. 282) is itself conspicuous.

Author Response

Q1: formulate a relevant research question, objective or hypothesis, design a study to address those, search and analyze appropriate data, and draw valid conclusions. Response: We rewrite the last part of Introduction to make clear of research question, objective or hypothesis, and study design. (From L80-96) Many previous studies discussed covariance exists between biodiversity and ecosystem services (ES), or between biodiversity and vegetation biomass. These researches provided valuable information of conservation planning, though few of them talked about scale differences of the relationships. Therefore, by this research, we want to answer the question that whether the ES and vegetation with high productivity can be covered by biodiversity conservation at fine and coarse scale? And which factors drive the biodiversity distributions, and therefore impact the relationship between biodiversity and the other two kinds of indicators. Details are as follows: The interrelationship between biodiversity and ecosystem function depends on factors such as ecosystem type, climate, topography, soil, and human disturbance, which are closely correlated with biomass and ecosystem productivity [17-18]. It is not suitable to use all the indicators during the planning stage, which easily results in overprotection and reduces the applicability of the conservation plan. If the spatial relationships among the factors can be clarified, they can provide an important reference for ecological conservation planning and help to gain the synergy between biodiversity and regulating ecosystem services. However, the relationships might vary by spatial scale, and there are few comparative studies on this regime. A measurement and discussion of this issue will benefits mountain nature reserve planning and setting priori conservation area at national conservation. In order to explore the covariance between RES, vegetation, and biodiversity at different scales and identify the scale differences at which spatial overlap acts and measure the contribution of determinants impacting biodiversity distributions at both scale, we performed the analysis based on mapping of water retention, soil retention (RES), NPP, biomass (ecosystem productivity), mammal, bird, plant, and sum of all indicator species (biodiversity). The covariance analysis begins by spatial intersection and correlation between biodiversity and RES, following by a spatial lag model testing the relationships between biodiversity and vegetation productivity. Furthermore, we employed a Boost Regression Tree model (BRT) to estimate biophysical factors driving biodiversity distributions at both scales. Therefore, the objectives of this study are (1) to measure the correlations of biodiversity, ecosystem functions, and ecosystem productivity and (2) to identify the key factors impacting the biodiversity distribution between scales. Q2: address biodiversity, indicators and other basic terms and concepts correctly. The sentence "Biodiversity is an important indicator for measuring the importance of natural ecosystems ..." (l 54-55) clearly shows a serious misunderstanding of key foundations. Your conclusions are not supported, by data, method or study design, or by a combination of these. Your main conclusions that mammal species richness is an indicator for conservation, that NNP and biomass influence biodiversity and ecological functions, and that scale is a dependent variable, are very superficial and can be found in any basic text book in biology and ecology. Also, other variables ("latent", l. 282) for sure are both numerous and very important to consider. The fact that you chosed the term "might"  (l. 282) is itself conspicuous. Response: "Biodiversity is an important indicator for measuring the importance of natural ecosystems ..." (l 51) have been revised to “Biodiversity maintenance is an important function of natural ecosystems”. This paragraph used to explain why we chose species richness to surrogate biodiversity, because the definition of biodiversity involves more complex contents. And we also deleted some unscientific expressions such as “might be”.   Conclusions have been rewritten to be more conclusive as follows (L275-282):   This study shows that the spatial scale of conservation planning should be taken into full account for the determination of ecological indicators for conservation planning. The spatial overlap and correlation analysis demonstrated that in a mountain region with a high value of biodiversity sustainability and regulating ecosystem services, mammal species richness is a core indicator for establishing a conservation network which are mainly driven by biophysical and climatic factors. While, at a coarser spatial scale, plant species richness becomes more important to benefit synergy of ecosystem functions. Because at national scale, NPP and ecoregion also have high influences in addition to biophysical and climatic factors on the pattern of species richness.

Reviewer 2 Report

The submitted work is quite interesting but has some experimental flaws.

On the use of MaxEnt, there is no information on how the authors use the beta multiplier, what type of dataset splitting has been used, how they approach any sampling bias that eventually occur in the data, what is the spatial resolution of the predictors and how they choose the final model. These all, affect the model overfitting. On the correlation analysis, i suggest to use VIF along with Pearson and use also expert;s opinion. I suggest to rewrite the section of that and rerun the analysis by using Wallace R package that includes several methods for model selection and evaluation.

Author Response

Q1:On the use of MaxEnt, there is no information on how the authors use the beta multiplier, what type of dataset splitting has been used, how they approach any sampling bias that eventually occur in the data, what is the spatial resolution of the predictors and how they choose the final model. These all, affect the model overfitting. (Maxent Model) Response: We add more detailed information on Maxent parameters part and sampling methods in as follows (L136-140): In potential range of each indicator species, which decided by local monitoring reports and expert opinion, we made trace point survey of species along 381 line transect keeping 1 transect within 5*5km2 area. Points were not recorded within 150m distance of other records of same species to reduce sampling bias. 75% percent of the points were randomly selected for developing the model, and 25% of the points were used for the model validation by ROC analysis (Appendix S3, Figure S1 to S7 in supporting material). The continuous results were changed to Boolean maps by a threshold searching method based on kappa analysis [23]. Spatial resolution of all feeding variables were resampled to 90m. Meanwhile, we added ROC validation on supporting material of mammal indicator species as an example. Q2: On the correlation analysis, i suggest to use VIF along with Pearson and use also expert’s opinion. I suggest to rewrite the section of that and rerun the analysis by using Wallace R package that includes several methods for model selection and evaluation. (Correlation analysis) Response: Thank you for the very scientific suggestion. Correlation analysis includes three parts in this study. The first is the correlation test between species richness (SR) and regulating ecosystem services (RES) by Pearson coefficient. The goal of this test is to measure the spatial relationships between two kinds of variable above, and followed by a spatial overlap analysis (Figure 2). We didn’t use the regression model because that there is no study demonstrate clear causal relationship exists between SR and RES. The second part of correlation analysis tested relationship between NPP, biomass and SR, RES variables. In this step, a forward stepwise regression was used to remove collinearity in variables. The third part was used to identify the contribution of impact factors driving SR distribution at both scale cases. According to the comment, we revised the analysis method. Before the running of BRT model, we tested VIF along with a Least Squares Regression, and removed non-significant variables for feeding BRT model (Line 164-165, Table 3).